# Recent Developments in Mycobacteria-Based Live Attenuated Vaccine Candidates for Tuberculosis

**DOI:** 10.3390/biomedicines10112749

**Published:** 2022-10-29

**Authors:** Mario Alberto Flores-Valdez, Andreas Kupz, Selvakumar Subbian

**Affiliations:** 1Biotecnología Médica y Farmacéutica, Centro de Investigación y Asistencia en Tecnología y Diseño del Estado de Jalisco, Av. Normalistas 800, Col. Colinas de la Normal, Guadalajara 44270, Mexico; 2Centre for Molecular Therapeutics, Australian Institute of Tropical Health and Medicine (AITHM), James Cook University, Townsville, QLD 4811, Australia; 3Public Health Research Institute (PHRI), New Jersey Medical School, Rutgers University, Newark, NJ 07103, USA

**Keywords:** *Mycobacterium*, BCG, preclinical models, latent tuberculosis, vaccine, Esx system, MTBVAC, HIV-TB

## Abstract

Vaccination is an excellent approach to stimulating the host immune response and reducing human morbidity and mortality against microbial infections, such as tuberculosis (TB). Bacillus Calmette–Guerin (BCG) is the most widely administered vaccine in the world and the only vaccine approved by the World Health Organization (WHO) to protect against TB. Although BCG confers “protective” immunity in children against the progression of *Mycobacterium tuberculosis* (Mtb) infection into active TB, this vaccine is ineffective in protecting adults with active TB manifestations, such as multiple-, extensive-, and total-drug-resistant (MDR/XDR/TDR) cases and the co-existence of TB with immune-compromising health conditions, such as HIV infection or diabetes. Moreover, BCG can cause disease in individuals with HIV infection or other immune compromises. Due to these limitations of BCG, novel strategies are urgently needed to improve global TB control measures. Since live vaccines elicit a broader immune response and do not require an adjuvant, developing recombinant BCG (rBCG) vaccine candidates have received significant attention as a potential replacement for the currently approved BCG vaccine for TB prevention. In this report, we aim to present the latest findings and outstanding questions that we consider worth investigating regarding novel mycobacteria-based live attenuated TB vaccine candidates. We also specifically discuss the important features of two key animal models, mice and rabbits, that are relevant to TB vaccine testing. Our review emphasizes that the development of vaccines that block the reactivation of latent Mtb infection (LTBI) into active TB would have a significant impact in reducing the spread and transmission of Mtb. The results and ideas discussed here are only based on reports from the last five years to keep the focus on recent developments.

## 1. Introduction

Despite the availability of effective antibiotics, treatment for active tuberculosis (TB) often has a poor prognosis. Suboptimal adherence to lengthy treatment regimens contributes to the emergence of drug-resistant TB, which poses a significant burden on the global economy and patients’ quality of life [1]. The emergence of multiple-, extensive-, and total-drug resistant (MDR/XDR/TDR) TB has added to the global public health threat posed by TB, as treatment outcomes in these cases are generally poor and prophylaxis for contacts is unavailable [1]. 

There were 10 million new cases and 1.5 million deaths due to TB in 2020, and it is estimated that roughly 85–90% of *Mycobacterium tuberculosis* (Mtb)-infected immune-competent individuals, which amounts to 1.8 billion people globally, develop asymptomatic latent TB infection (LTBI). People with LTBI can reactivate with immune compromise and are thus a vast reservoir of potential new transmission sources upon reactivation [2]. This situation is exacerbated by HIV co-infection, which increases TB susceptibility, mortality, and the risk of LTBI reactivation. Hence, it is crucial to explore new avenues that may result in alternative tools, other than additional drugs, to control the progression of Mtb infection into active TB disease, such as by promoting enhanced immune responses early after primary infection to clear Mtb or by reducing the pathological damage associated with primary TB. Furthermore, developing vaccines that block the recurrence of LTBI into active TB would significantly reduce the spread of Mtb strains in the community. 

The primary strategy for new TB vaccines under development is based on data from human and animal studies, which suggest that the activation of Mtb-specific CD4^+^ T cells via immunodominant Mtb antigens confers adequate protection against TB. However, most new anti-TB vaccine candidates employing this strategy have not shown superiority over BCG, which includes, among other drawbacks and limitations, the risk of causing disease in individuals with HIV infection or other immune compromises, highlighting the need to identify novel approaches and/or to reinterpret existing concepts to guide the rational design of new TB vaccines [3,4,5]. In addition, the product profile of an ideal therapeutic vaccine for TB is unknown. Recombinant BCG strains have received significant attention as potential replacement vaccines for BCG [6,7,8]. In contrast to subunit vaccines, live attenuated mycobacterial vaccines elicit a broader immune response and usually do not require an adjuvant. 

Animal models are excellent tools to study the course of Mtb infection, including the pathological manifestations of acute and chronic active TB, as well as LTBI and reactivation, and to decipher the host and pathogen determinant(s) that are vital for pathogenesis. Knowledge gained from such studies can aid understanding of the pathogenesis of various forms of TB and the development of more effective intervention strategies to combat the disease. Several animal models, ranging from zebrafish to non-human primates (NHPs), have been explored to investigate their resemblance of biological manifestations during active TB and LTBI with corresponding human conditions [9]. Each animal model mimics some common clinical and pathological features of pulmonary TB in human patients, including pneumonia, granulomatous response, and cellular immunity processes such as cytokine/chemokine production [9]. However, some animal models, such as rabbits and NHPs, display more intricate host responses to Mtb infection, such as a heterogeneity in granulomatous response (comprising necrotic and caseating cavitary lesions) and the spontaneous establishment of LTBI that reactivates to active disease following immune suppression treatment [9]. Nonetheless, each animal model of TB has its advantages and limitations in replicating the complete spectrum of clinical disease features seen in human TB cases.

This article discusses the recent developments in novel recombinant BCG and Mtb-based vaccines and their pros and cons over conventional BCG vaccines that the World Health Organization currently approves for use in humans. We also highlight some novel mouse and rabbit models of TB as preclinical tools to screen potential TB vaccines.

### 1.1. rBCG Vaccines

The current TB vaccine, BCG, neither sterilizes Mtb infection nor effectively protects against pulmonary cavitary disease and the spread of infection in the human adult population, contributing the most to TB transmission in the community. Thus, the development of next-generation TB vaccines is of great urgency. In this regard, it was recently discussed that the development of efficacious rBCGs aimed at improving protection against TB often follows two strategies: (i) deleting genes from (knockouts) BCG or (ii) inserting mycobacterial antigen-coding genes into the BCG genome (knock-ins, KI) [7]. For the former category, VPM1002 (BCGΔ*ureC*::*hly*) is the most advanced vaccine candidate, which is based on BCG Danish. In a phase 2 clinical trial performed in South African newborn babies, VPM1002 was found to be better tolerated than BCG regarding side effects. These included 15-fold less grade 3–4 vaccine-related adverse reactions or lymphadenopathy of 10 mm or greater in diameter, 3-fold less scarring, and 10-fold less abscess formation in those who received VPM1002 compared with the BCG-administered group; however, ex vivo immune responses were higher in the group that received the BCG vaccine [10]. Whether reduced ex vivo immunogenicity is linked to a reduced protective efficacy of VPM1002 remains to be determined. A currently in-progress phase III study (*ClinicalTrials.gov Identifier: NCT04351685*) will inform about the immunogenicity afforded by VPM1002. Additionally, compared with wild-type BCG Danish, BCGΔ*ureC*::*hly*Δ*nuoG* (a VPM1002 variant with the deletion of the *nuoG* gene) reduced the replication of a laboratory strain (H37Rv) as well as a clinical Mtb strain (W/Beijing) in the lungs and spleens of infected mice, especially during chronic TB [11]. However, the translation ability and further development of this new vaccine derivate against human TB remain to be determined. 

Another BCG-based vaccine candidate that follows the gene-deletion strategy is BCGΔBCG1419c (based on BCG Pasteur and devoid of the c-di-GMP phosphodiesterase-encoding gene *BCG1419c*). Compared with parental BCG, BCGΔBCG1419c induced different levels of TNF-α, IL-6, and IL-1β; was attenuated for intracellular replication in murine macrophages; and changed the expression of antigenic proteins when grown as surface pellicles [12]. We later observed that compared with parental BCG Pasteur, the BCGΔBCG1419c reduced lung pathology and IL-6 levels; furthermore, only BCGΔBCG1419c-vaccinated C57BL/6 mice showed a reduction in the production of TNF-α and IL-10 compared with non-vaccinated mice after six months of Mtb H37Rv infection [13]. We further evaluated the differences in the antigenic repertoires of BCGΔBCG1419c and BCG at the transcriptional level, which was later extended to the proteome level in cellular and secreted proteins [14]. In another study, we observed a superior capacity of BCGΔBCG1419c to induce memory T cells following intratracheal vaccination compared with its parental BCG, as well as very low numbers of detectable Mtb in the spleen and lungs, particularly at 120 days post-infection [15]. The increased induction of polyfunctional central and effector memory CD8^+^ T cells with a similar CD4^+^ T cell immunogenicity was observed in C57BL/6 mice vaccinated with BCGΔBCG1419c compared with those that received BCG Pasteur; this differential immunogenicity also resulted in reduced pulmonary inflammation in the former group following infection with the Mtb M2 clinical strain [16]. We further showed that BCGΔBCG1419c-immunized mice induced longer-lasting immune responses, with reduced frequencies of exhausted CD4^+^ T helper (TH) cells and increased frequencies of IL10-producing T cells [17].

Finally, regarding the host-protective efficacy of the BCGΔBCG1419c vaccine in murine models, we found that it contributed to reduced pneumonia and alveolitis in a chronic TB type 2 diabetes model via different modulations of cellular responses. However, it only provided transient protection in a hypersusceptible I/St mice model of Mtb infection [18,19]. In light of the latter results, we would like to mention the need to test novel TB vaccine candidates in more than one mouse strain, as the heterogeneity of protection following BCG vaccination is host-dependent; for instance, diverse outbred mice may show differential immune activation and protection against subsequent Mtb infection [20]. Though the current recommendation for TB vaccine development considers evaluating efficacy in a second animal model, we think that additional mice strains should also be considered in this policy [21]. In accordance with these recommendations, we recently reported that vaccination with BCGΔBCG1419c protects against pulmonary and extrapulmonary TB and is safer than BCG in a guinea pig model of Mtb infection [22]. Together, the results from two preclinical animal models provide compelling evidence to further develop the BCGΔBCG1419c vaccine for human use.

In contrast to BCGΔBCG1419c, a modified vaccine candidate constructed by gene-insertion, rather than gene-deletion, is the BCG::phoPR strain, in which the *phoPR* genes from BCG Pasteur were inserted into BCG Japan (to restore it from its natural point mutations in these genes). Vaccination with BCG::phoPR was found to prolong the survival of Mtb-infected guinea pigs compared with those vaccinated with the parental BCG [23]. However, it is presently unknown whether BCG::phoPR will be further developed as a potential vaccine candidate or instead serve as a basis for a possible restoration of gene function in other rBCG vaccine candidates.

### 1.2. rBCG with ESX System

The modified BCG strain containing the “ESAT6 protein family secretion system-1” from Mtb (BCG::ESX1^Mtb^, also known as BCG::RD1) is amongst the most effective rBCG strains ever developed, at least at the preclinical level. Although BCG::ESX1^Mtb^ reduces the bacterial burden and is more immunogenic than BCG in animal models, the strain has been deemed unsafe due to its prolonged persistence and increased virulence in immunocompromised hosts [24]. The increased protection of BCG::ESX1^Mtb^ was mainly attributed to an improved T cell response to ESAT6 (also known as EsxA), an immunodominant protein secreted by the ESX1 locus of Mtb [24]. However, more recent studies have identified that full-length ESAT6 secretion is required to induce cytosolic pattern recognition and protective non-cognate immune responses against Mtb [25,26]. Importantly, in a mouse model of TB, when using 24 vaccine regimens consisting of three BCG strains (BCG, BCG::ESX1^Mtb^, and a derivative of BCG::ESX1^Mtb^) and eight combinations of vaccine delivery, we recently demonstrated that the inclusion of ESX1-exported effector molecules was the overall most efficient way to improve BCG performance [27]. Furthermore, we showed that the mucosal delivery of ESX1-expressing BCG strains also led to near-sterilizing immunity against Mtb in a mouse model of diet-induced type 2 diabetes [28]. These studies collectively provided the rationale to develop rBCGs with Mtb ESX1-like proteins, in which ESAT6 secretion has been uncoupled from the detrimental effects of the ESX1 secretion system for better protection against TB. Accordingly, Dr. Kupz generated an rBCG strain, BCG::ESAT6-PE25SS, by fusing the Mtb *esxA* gene to the general secretion signal of the mycobacterial type VII secretion pathway [29]. This new rBCG secretes full-length ESAT6 via the ESX5 secretion system that, in contrast to ESX1, is also present in BCG [30]. The ESX5-mediated delivery of ESAT6 combines several advantages, such as cytosolic contact, the induction of ESAT6-specific T cells, and protection against TB, with low/mild virulence and reduced persistence in immunocompetent and immunocompromised animal hosts. Additionally, compared with that of BCG::ESX1^Mtb^ and parental BCG, the mucosal administration of BCG::ESAT6-PE25SS is associated with a more rapid clearance of Mtb from the infected mouse lungs [30,31]. This suggests that the further evaluation of BCG::ESAT6-PE25SS in a second preclinical model and the consideration of making additional genetic modifications might improve the immunogenicity of this vaccine candidate.

Another rBCG that follows the rationale of using RD1, from *M. marinum* in this instance, is BCG::ESX1^Mmar^. This vaccine candidate significantly reduced Mtb HN878 and M2 loads in the lungs and spleens of infected C57BL/6 mice compared with those vaccinated with parental BCG. However, no significant decrease in pulmonary inflammation was noted between vaccinated and non-vaccinated Mtb-infected animals [32]. 

### 1.3. MTBVAC

Another mycobacteria-based live attenuated TB vaccine candidate is MTBVAC, developed from the Mt103 clinical isolate (belonging to lineage 4-L4-) and devoid of the *fadD26* and *phoP* genes. During preclinical studies of MTBVAC, C3H/HeNRj mice were chosen for further studies because a host-dependent effect was noted in comparison with BALB/c and C57BL/6 mice, as MTBVAC afforded protection similar to that of BCG Pasteur in the latter two mouse strains [33]. Considering the possible impact of bacterial genetics on vaccine efficacy, different versions of MTBVAC based on Mtb isolates from L2 and L3 were recently produced and evaluated for their efficacy in C3H/HeNRj mice. In this study, the original MTBVAC (L4) had equal efficacy to BCG Pasteur against Mtb H37Rv (L4) and the EAI strain HCU3524 (L3), and it showed an improved reduction in lung CFU against the Mtb W4-Beijing strain (L2) at only four weeks post-intranasal challenge [34]. The L2- and L3-based versions of MTBVAC showed a similar capacity to reduce Mtb burden in the lungs and spleens of infected mice compared to L4-MTBVAC and BCG Pasteur, except that L3-MTBVAC was less able to reduce the replication of the Mtb W4-Beijing strain in the spleens of infected mice [34]. 

Recently, two genes, *cpnB* or *disA*, involved in cyclic-di-adenosine monophosphate (c-di-AMP) metabolism were deleted from MTBVAC to produce MTBVACΔcnpB::Km and MTBVACΔdisA::Km [35]. It was found that while the deletion of *cnpB* attenuated MTBVAC, the deletion of *disA* had no major effect on virulence in the immunized SCID mice [35]. Furthermore, the vaccination of C3H/HeNRj mice with MTBVACΔcnpB::Km or MTBVACΔdisA::Km afforded a similar level of protection as that conferred by MTBVAC and parental BCG Pasteur against short-term infection with Mtb Beijing W4. Together, these findings suggest that c-di-AMP may not be a good antigen for vaccine development against TB, at least in the context of the experimental conditions used in this study [35].

In a separate study performed in an advanced preclinical NHP model, rhesus macaques vaccinated with BCG or MTBVAC and challenged with an ultra-low dose of Mtb Erdman showed no difference in disease progression. However, at 16 weeks post-infection, the MTBVAC-vaccinated animals outperformed the BCG-vaccinated ones in reducing lung pathology and extrapulmonary bacterial loads [36]. It should be noted that depending on the host species (mouse versus NHPs) and the characteristic of the Mtb strain used for the infection, the superior protection (or not) of MTBVAC compared with BCG was found, therefore raising the question on the usage of specific animal models for vaccine testing and how general or specific the application of MTBVAC (or, for this purpose, any other TB vaccine candidate) would need to be to effectively prevent or reduce TB burdens in humans. Finally, MTBVAC has shown acceptable reactogenicity and induced a durable CD4 cell response in vaccinated South African infants, and additional clinical trials are in progress (*Clinical trial: NCT03536117*). The advantages and limitations of some mycobacteria-based novel vaccine candidates are summarized in Table 1.

### 1.4. A Novel Mouse Model of Latent Mtb Infection and Reactivation of TB in the Context of HIV-TB Co-Infection

Progressive CD4^+^ T cell depletion has been considered the primary cause of the HIV-induced reactivation of LTBI, and this complex interaction was recently modeled in NHPs [61]. Despite its close resemblance to the LTBI and reactivation occurring in humans, the use of this model is not widely available, therefore limiting the capacity of testing all vaccine candidates aiming to reduce the burden of LTBI. Therefore, in 2016, a mouse model used to study the reactivation dynamics of LTBI following immunosuppression through the loss of CD4^+^ T cells was reported [62]. In this model, intradermally (i.d.) infected C57BL/6 mice were found to contain Mtb within the local draining lymph nodes, but following the depletion of CD4^+^ cells, the mice underwent the progressive, systemic spread of the bacteria accompanied by lung pathology. This model was used to determine that vaccination with BCGΔBCG1419c or BCG Pasteur prevented the systemic spread of Mtb from the infected lymph node to the spleen and lungs, which was a measure of the reactivation of LTBI, thus showing that lympho-centric LTBI mice could be a tractable and reproducible small animal model with potential application for TB vaccine testing [15]. The model may also reveal additional immune cell populations that can be selectively targeted for the vaccine-mediated prevention of LTBI reactivation.

### 1.5. Evaluation of BCG Vaccine Efficacy in a Rabbit Model of Active Pulmonary TB

Selecting a suitable small animal model that can develop cavitary TB, as well as spontaneous LTBI and the immunosuppression-mediated reactivation of LTBI, is crucial for evaluating novel therapeutic and prophylactic TB vaccine candidates. The rabbit model of pulmonary Mtb infection produces the range of pathologic manifestations seen in patients with asymptomatic and symptomatic cavitary TB [63,64]. In the rabbit model of active pulmonary TB, we evaluated the protective efficacy of BCG against infection with ultra-low (<5 CFU), low (10–100 CFU), and high (100–500 CFU) Mtb inoculum doses [65]. We found that the prior vaccination of rabbits with BCG significantly reduced the lung bacterial load in Mtb-infected animals, though only at the ultra-low dose. However, at a higher infectious inoculum, Mtb CFU were insignificantly reduced in the BCG-vaccinated animals compared to the non-vaccinated group. Thus, the protective immunity conferred by BCG seems to be negatively impacted by the initial infectious bacterial load implanted in the lungs. At a very low infectious Mtb load, the host immunity against infection itself is sufficient to contain the progression of the infection into active disease [65]. In this setting, BCG vaccination appears to enhance and augment the host immunity in preventing disease progression. However, when the infectious inoculum implanted in the lungs is high, the infection can rapidly progress into active disease despite the onset of a host-plus BCG-induced, host-protective immunity. Therefore, there appears to be a delicate balance in the immune responses elicited in the host between Mtb infection and BCG vaccination. However, the overlap of immune components between host immunity induced following Mtb infection, those elicited by the BCG vaccine, and their contribution to the net outcome in protecting the host against infection remains to be determined

In a separate study, our group compared the response of BCG-vaccinated rabbits following central nervous system (CNS) infection with the Mtb strains HN878 or H37Rv [66]. In this study, BCG vaccination showed a significantly less protective efficacy in animals challenged with the more virulent Mtb HN878 strain compared with the laboratory Mtb strain, H37Rv, as indicated by more severe CSF inflammation, body weight loss, and bacillary dissemination to the liver and spleen in the former group of animals [66]. These results demonstrated that the extent of protective immunity conferred by the BCG vaccine depends on the nature and virulence potential of the infecting Mtb strain used in the challenge. Furthermore, the differences in disease pathology, lung bacillary load, and transcriptome profile between vaccinated and non-vaccinated animals were found to be quantifiable and reproducible in the rabbit model of cavitary TB [63,64,65]. These readouts can help characterize the immune response of various TB vaccines in the rabbit model.

Recently, several molecular tools have been developed to characterize the host immune responses in rabbits, including RNASeq analysis to decipher the genome-wide transcriptome profiling, functional classification, and the pathway analysis of differentially expressed genes in the granulomas; flow-cytometry methods have been established with additional antibodies for rabbits to create better immune cell phenotyping and functional assays than before. Furthermore, we developed an mRNA-based fluorescent in situ hybridization (mRNA-FISH) technique to map the spatial transcriptome of immune cells. In this technique, multiple gene transcripts can be probed at the site of infection (e.g., lung granulomas) using a combination of multiple mRNA targets tagged with different fluorescent markers [65,67,68].

## 2. Summary and Conclusions

Considering that most novel TB vaccine candidates have been tested against the laboratory Mtb H37Rv strain challenge and their efficacy has mostly been evaluated in murine models of active TB post-infection in the short-term, we think that to produce more effective TB vaccines, it is worth addressing the following questions: (a) What is the efficacy of novel (and those currently in clinical trials) vaccine candidates against multidrug-resistance (MDR) Mtb strains and how do these vaccine candidates work in adjunct antibiotic therapy? (b) Would they effectively reduce the TB caused by clinical Mtb isolates of variable virulence potential? (c) Would the routes of vaccine administration (for example: aerosol, intranasal, intratracheal, or intravenous) impact the immunogenicity and protective efficacy of a given vaccine? (d) What is the comparative profile of the immunogenicity and protective efficacy conferred by the vaccine candidates in various preclinical animal models, including mice, guinea pigs, rabbits, ferrets, and NHPs? (e) What is the contribution of host genetics, sex, age, and other comorbidities to the efficacy of these vaccine candidates? (f) Would there be any benefit of combining multiple live attenuated mycobacteria-based vaccine candidates with subunit (based on proteins, nucleic acids, or viral-vectors) vaccine candidates in terms of preventing or reducing TB? (g) Would strategies based on nanotechnology improve TB vaccine efficacy?

In addition, it is crucial to explore new avenues that may result in alternative tools, other than additional drugs, to control the progression of Mtb infection into active TB disease, such as by promoting enhanced immune responses soon after primary infection to clear Mtb or reducing the pathological damage associated with primary TB. Furthermore, developing vaccines that block the recurrence of LTBI into active TB would significantly reduce the spread of Mtb strains in the community.

In conclusion, novel and innovative vaccine candidates based on BCG and Mtb strains are emerging with superior safety, immunogenicity, and protective efficacy against TB compared with parental strains. Since these vaccine candidates are currently at various developmental stages, their intricate mechanisms of action are not fully understood and warrant more extensive in vitro, in vivo, ex vivo, and clinical evaluation studies. Such future studies should inform the suitability of these potential vaccine candidates to prevent TB in humans.

## Figures and Tables

**Table 1 biomedicines-10-02749-t001:** Summary of various mycobacteria-based vaccine candidates and their properties.

Mycobacterial Strain/Modification Strategy	Findings	Limitations	References
**Inclusion of ESX-1**			
BCG::ESX-1^Mtb^ (BCG::RD1)	Reduced the bacterial burden, and it was more immunogenic than BCG in animal models.	It was deemed unsafe due to its prolonged persistence and increased virulence in immunocompromised hosts.	[34,37]
BCG::ESX-1^Mtb^-ESAT6 L28A/L29S	Showed strong attenuation in mice and a high protective efficacy both in mouse and guinea pig models. Safer than parental BCG::ESX-1^Mtb^ in immunocompromised hosts.	It may give false positive results in current IFN-g release assays (e.g., Quantiferon^®^) because of the presence of ESAT-6.	[38]
BCG::ESX-1^Mmar^	Reduced Mtb HN878 and M2 loads in the lungs and spleens of C57BL/6 mice compared with parental BCG.	It did not improve lung pathology in murine models. Further characterization in additional models remains to be seen.	[31]
**Secretion of *Mtb* antigens**			
rBCG30	Induced greater protection than BCG against aerosolized Mtb Erdman in multiple animal models in different labs. Human Phase 1 study showed that rBCG30 is safe and induces significantly enhanced antigen-specific immune responses.	Further characterization in additional models remains to be seen.	[39,40,41,42]
rBCG30(*mbtB*)	Could not cause disseminated BCGosis. Safe and more potent than BCG against aerosolized Mtb Erdman in animal studies and safer than BCG in SCID mice.	The need to add iron during manufacture could pose technical difficulties in the yield and/or viability of the vaccine.	[43]
rBCG E6	Induced a mixed Th1/Th2 response in mice.	It may give false positive results in current IFN-γ release assays (e.g., Quantiferon^®^) because of the presence of ESAT-6.	[44]
rBCG-1173:A	Improved efficacy compared with BCG in mice at 4 weeks post-infection with *M. tuberculosis* H37Rv.	Further characterization in additional models remains to be seen. It contains Ag85A, which is part of several other vaccine candidates.	[45]
AERAS-401	A good safety profile in SCID mice.	Comparable protection to that afforded by BCG in mice.	[46,47]
AERAS-422	Immunogenic and safe in mice and guinea pigs. Increased protection against Mtb HN878 challenge in mice. Very safe in SCID mice.	Not as effective in a long-term survival assay with clinical Mtb isolates compared with BCG Pasteur. The unexpected reactivation of varicella zoster virus (shingles) in two of eight healthy adult vaccine recipients resulted in the discontinuation of the AERAS-422 vaccine development.	[48,49]
(H)PE-ΔMPT64-BCG	Reduced *M. tuberculosis* Erdman bacterial burden in the lungs and spleen of C57BL/6 mice relative to BCG. Reduced *M. tuberculosis* bacterial burden in spleen relative to BCG in guinea pigs, with no improvement for lung *M. tuberculosis* loads.	Further characterization in additional models remains to be seen.	[50]
BCG::ESAT6-PE25SS	Mucosal administration of BCG::ESAT6-PE25SS was associated with the more rapid clearance of *M. tuberculosis* H37Rv from infected mouse lungs compared with BCG and BCG::ESX1^Mtb^.	Further characterization in additional models remains to be seen.	[32]
**Increased phagolysosome delivery**			
BCG Δ*zmp1*	A slight reduction in lung bacterial loads compared with BCG in guinea pigs. Increased immunogenicity compared with BCG. Showed a better safety profile in SCID than the parental BCG strain.	Further characterization in additional models remains to be seen.	[51]
BCG Δ*ureC::hly* (VPM1002)	Superior efficacy compared with parental BCG in murine models, somewhat improved safety, and reduced ex vivo immunogenicity in human clinical trials compared with BCG.	Unknown efficacy against TB in susceptible hosts (preclinical); efficacy in humans to be determined in Phase III trials.	[10,52,53]
BCGΔ*ureC::hlyΔnuoG*	Reduced replication in the lungs and spleens of *M. tuberculosis* H37Rv or W/Beijing-infected mice compared with wild-type BCG Danish, especially during chronic TB.	Further characterization in additional models remains to be seen.	[11]
**Latency and pathogen resistance**			
rBCG:XB	Significant decrease in bacterial burden in C57BL/6 mice intranasally infected with *M. tuberculosis* H37Rv at 4 and 20 weeks post-infection.	Further characterization in additional models remains to be seen.	[54]
rBCG-LTAK63lo	BALB/c mice immunized with rBCG-LTAK63lo had reduced bacterial loads at 4 weeks and 4 months post-infection after intratracheal challenge with *M. tuberculosis* H37Rv and after 4 weeks post-infection with *M. tuberculosis* Beijing 1471.	Further characterization in additional models remains to be seen.	[55,56]
BCG::dosR	Led to the induction of most of the DosR-regulon genes without the need to use hypoxic cultures.	No information about immunogenicity or efficacy reported.	[57]
BCG::phoPR	Prolonged the survival of guinea pigs infected with *M. tuberculosis* H37Rv compared with parental BCG.	Further characterization in additional models remains to be seen.	[58]
**Modification of Cyclic dinucleotide metabolism**			
BCGΔBCG1419c	Increased in vitro biofilm formation compared with BCG and expected to produce increased levels of cyclic di-GMP (c-di-GMP). Improved control of chronic TB in murine models, active pulmonary TB, and extrapulmonary TB in guinea pigs compared with parental BCG. Improved control of chronic TB in a type 2 diabetes murine model. Increased the induction of effector and central memory T CD8^+^ cells compared with BCG, with reduced pulmonary inflammation following *M. tuberculosis* M2 infection.	Further characterization in advanced preclinical models is needed.	[12,13,16,18,59]
BCG::disA	Overexpressed the cyclic diadenylate cyclase gene (*Rv3586*) in BCG Pasteur. Significantly reduced *M. tuberculosis* H37Rv-driven lung pathology at 14 weeks post-infection with a control of bacterial burden similar to that of BCG; significantly reduced *M. tuberculosis* H37Rv burden at 18 weeks post-infection with a control of lung pathology similar to that of BCG.	Further characterization in advanced preclinical models is needed.	[60]
**MTBVACΔ *cnpB* and MTBVACΔ*disA***	Deletion of *cnpB* attenuated MTBVAC, while the deletion of *disA* had no significant effect on virulence in SCID mice.	Deleting genes involved in c-di-AMP metabolism did not change the protective efficacy compared with the parental MTBVAC against short-term infection with Mtb Beijing W4 in C3H/HeNRj mice and conferred the same protection as that elicited by BCG Pasteur. Further characterization in advanced preclinical models is needed.	[35]

## Data Availability

Not applicable.

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
