# Peer review of "Recent Developments in Mycobacteria-Based Live Attenuated Vaccine Candidates for Tuberculosis"

_biomedicines, 2022, doi:10.3390/biomedicines10112749_

Round 1

Reviewer 1 Report

In this review article “Recent developments in mycobacteria-based live attenuated

vaccine candidates for tuberculosis” authors have provided significant insight into its mode of action of mycobacteria-based live attenuated vaccine candidates for tuberculosis. The content of the manuscript is informative but unfortunately the content is not well written and presented, and required modifications with addition of recent research articles. 

The manuscript is suitable for publication after major revision.

Comments:

Firstly, a very closely related study has been published earlier, hence the novelty of the submitted work is low.

To simplify the article's content, the authors must include a table and draw images for better understanding. 

I suggest the improvements as a summary with the following sections:

1. More details animal models relevant to TB vaccine testing 

2. Suggest improving the conclusions and future perspectives mentioning (venturing thoughts) which will be the possible clinical improvements by this futuristic approach. 

3. Additionally, taking into account the focus of this review is the antimicrobial activity of these vaccine, the authors should emphasize this important property for selected compounds in the concluding remarks item and elaborate. 

Author Response

Comment: Firstly, a very closely related study has been published earlier, hence the novelty of the submitted work is low.

Response: We agree that there are several review articles published recently in this subject matter (e.g: PMID:36091032; 30000153; 35967410; 35891320;35812462; 35632558).  These reviews highlight the history, pros and cons of BCG and general animal models used for vaccine testing. However, our report is unique and novel in two concepts:1). The description about a novel, lympho-centric mice model of Mtb infection that results in latency and reactivation, similar to HIV-TB co-infection in humans; 2). The rabbit model of Mtb infection and its relevance to vaccine testing. In addition, we focused on the most recent developments of recombinant BCG and Mtb-based vaccines that have the potential to be tested in humans. Therefore, we believe that our report has a unique and added value to the literature on TB vaccines.

Comment: To simplify the article's content, the authors must include a table and draw images for better understanding. 

Response: As suggested by the reviewer, we have added a new table to summarize the various rBCG/mycobacteria-based vaccines, their advantages, and limitations and cited relevant references.

Comment: 1. More details on animal models relevant to TB vaccine testing 

Response: As the other reviewer (Reviewer#1) has pointed out above (see comment#1), there are recently published reviews that have provided a comprehensive report on various animal models of TB.  The main focus of this article is not on summarizing various animal models relevant to TB vaccine testing; however, we have provided novel information on two very crucial models (see our response above) at a high-level and summarized some of the key information in this article.

Comment: 2. Suggest improving the conclusions and future perspectives mentioning (venturing thoughts) which will be the possible clinical improvements by this futuristic approach. 

Response: As suggested by the reviewer, we have updated the summary, conclusion and future perspective, relevant to the focus of our article.

Comment: 3. Additionally, taking into account the focus of this review is the antimicrobial activity of these vaccine, the authors should emphasize this important property for selected compounds in the concluding remarks item and elaborate.

Response: Though our review is focused on the host-beneficial effects of mycobacteria-based vaccines, the mechanism of action of most of these new family of vaccines are largely unknown.  As we mentioned in the review, these are new generation vaccines that have several questions, such as the antimicrobial properties etc., unanswered. Nonetheless, we have updated the summary, conclusion and future perspective as relevant to the focus of our article.

Reviewer 2 Report

In this paper, the authors present a brief review of the recent advances in mycobacteria-based live vaccines for the control of tuberculosis (TB). The work is relevant considering the recent increase in the number of TB cases worldwide and also the need for effective and novel vaccines against the disease in humans and animals. As mentioned in the title, the manuscript is focused on mycobacteria live attenuated vaccines against TB; however, several sections of the document are dedicated to animal models instead of vaccine efficacy and protective immune responses elicited by the anti-TB experimental live attenuated vaccines. In addition, some sections are missing conclusions and wrap-up sentences. Below are comments.

- The abstract section needs a brief conclusion based on the reviewed information presented in the study.

- Lines 54-64: This paragraph is about vaccines, and it should be expanded. In fact, I miss more information on BCG, and this especially important considering that BCG is the only attenuated strain used as an anti-TB vaccine to date. Therefore, it would be great to have more information, such as a brief history of BCG, its cons and pros and then introduce its use as a recombinant vaccine against TB.

- Lines 66 to 82, 225 to 240, and 242 to 248: These sections are too concentrated on animal models. The authors should focus on the mycobacterial vaccines and the immune responses elicited by these vaccines.

- Lines 95 to 100: BCG is well known for its safety. Therefore, the authors should explain the comparison of wtBCG and VPM1002 in “terms of side effects” in newborn babies.

- Lines 205 to 210: This paragraph is too long and difficult to follow. The authors need to rephrase this paragraph, and perhaps break it into two or three sentences.

- A conclusion on mycobacteria-based live vaccines is missing from the “Summary and conclusions” section. The authors need to address this issue.

Minor comments:

- Line 26: I suggest the use of “wildtype” instead of “conventional” BCG.

- Line 20: there is an extra-space in “total-drug”.

- Line 36: spell out “TB” in the first time that it is written and then use the acronym afterwards.

- Lines 38 and 39: the authors need to spell out MDR and MDR/XDR/TDR and then use the acronym afterwards.

- Line 43: spell out “Mtb”.

- Line 227: NHP instead of “non-human primate”.

- Line 265: the authors need to review and rephrase the statement “disease pathology”.

- Line 295: “Grant number??”

- I suggest that authors should revise the entire document for acronyms citations. Please follow the rule of spelling out the concept in the first time that it is written and then use the acronym afterwards.

Author Response

Comment: - The abstract section needs a brief conclusion based on the reviewed information presented in the study.

Response: As suggested by the reviewer, we have included a conclusion statement in the abstract section of our article.

Comment: - Lines 54-64: This paragraph is about vaccines, and it should be expanded. In fact, I miss more information on BCG, and this especially important considering that BCG is the only attenuated strain used as an anti-TB vaccine to date. Therefore, it would be great to have more information, such as a brief history of BCG, its cons and pros and then introduce its use as a recombinant vaccine against TB.

Response: As pointed out by Reviewer#1, there are several, more elaborate review articles published recently about BCG, its history, pros and cons (e.g: PMID:36091032; 30000153; 35967410; 35891320;35812462; 35632558).  Therefore, this aspect is not the focus of our current review. Rather, our report is unique and novel in two concepts:1). The description of a novel, lympho-centric mice model of Mtb infection that results in latency and reactivation, similar to HIV-TB co-infection in humans; 2). The rabbit model of Mtb infection and its relevance to vaccine testing. In addition, we focused on the most recent developments of recombinant BCG and Mtb-based vaccines that have the potential to be tested in humans. Therefore, we believe that our report has a unique and added value to the literature on TB vaccines.

Comment:- Lines 66 to 82, 225 to 240, and 242 to 248: These sections are too concentrated on animal models. The authors should focus on the mycobacterial vaccines and the immune responses elicited by these vaccines.

Response: We observed that this comment contrasts with what Reviewer-1 has asked for. Therefore, we have taken a balanced approach to highlight only the most recent developments in the TB vaccine field and selected preclinical animal model systems that are relevant to test these vaccines.

Comment:- Lines 95 to 100: BCG is well known for its safety. Therefore, the authors should explain the comparison of wtBCG and VPM1002 in “terms of side effects” in newborn babies.

Response: As suggested by the reviewer, we have summarized the comparison of wtBCG and VPM1002 in terms of their side effects in newborn babies, and cited relevant references in the revised version.

Comment:- Lines 205 to 210: This paragraph is too long and difficult to follow. The authors need to rephrase this paragraph, and perhaps break it into two or three sentences.

Response: As suggested by the reviewer, we have rephrased and updated this section in the revised version.

Comment:- A conclusion on mycobacteria-based live vaccines is missing from the “Summary and conclusions” section. The authors need to address this issue.

Response: As suggested by the reviewer, we have added a conclusion statement in the revised version.

Minor comments:

Comment:- Line 26: I suggest the use of “wildtype” instead of “conventional” BCG.

Response: We modified this word; however, mentioning “wildtype” may lead to confusions to many microbiologists.

Comment:- Line 20: there is an extra-space in “total-drug”.

Response: Corrected.

Comment:- Line 36: spell out “TB” in the first time that it is written and then use the acronym afterwards.

Response: Corrected.

Comment:- Lines 38 and 39: the authors need to spell out MDR and MDR/XDR/TDR and then use the acronym afterwards.

Response: Corrected.

Comment:- Line 43: spell out “Mtb”.

Response: Corrected.

Comment:- Line 227: NHP instead of “non-human primate”.

Response: Corrected.

Comment:- Line 265: the authors need to review and rephrase the statement “disease pathology”.

Response: “disease pathology” is the correct phrase, which has been used in published work and as such accurately reflects the terminology required for this sentence. Kolloli, A.; Kumar, R.; Singh, P.; Narang, A.; Kaplan, G.; Sigal, A.; Subbian, S. Aggregation state of Mycobacterium tuberculosis impacts host immunity and augments pulmonary disease pathology. Commun Biol 2021, 4, 1256, doi:10.1038/s42003-021-02769-9.

Comment:- Line 295: “Grant number??”

Response: Added.

Comment:- I suggest that authors should revise the entire document for acronyms citations. Please follow the rule of spelling out the concept in the first time that it is written and then use the acronym afterward

Response: As suggested by the reviewer, we have revised the entire article for acronyms and updated it accordingly.

Round 2

Reviewer 1 Report

- The revision is satisfactory, and authors rectified the sections as I suggested earlier. Hence, I recommended for publication. 

- I would like to suggest the authors to include the abbreviation list (all mentioned one) at the end of this manuscript.

Reviewer 2 Report

The authors have satisfactorily addressed my comments.